# Immune Therapy for Liver Cancers

**DOI:** 10.3390/cancers12010077

**Published:** 2019-12-27

**Authors:** Marc Hilmi, Angélique Vienot, Benoît Rousseau, Cindy Neuzillet

**Affiliations:** 1Department of Medical Oncology, Curie Institute, University of Versailles Saint-Quentin, 35 rue Dailly, 92210 Saint-Cloud, France; Marc.hilmi@gustaveroussy.fr; 2GERCOR Group, 151 rue du Faubourg Saint-Antoine, 75011 Paris, France; angelique.vienot@inserm.fr (A.V.); rousseab@mskcc.org (B.R.); 3Department of Medical Oncology, Besançon University Hospital, 3 Boulevard Alexandre Fleming, 25030 Besançon, France; 4Department of Medicine, Division of Solid Tumor Oncology, Memorial Sloan Kettering Cancer Center, New York, NY 10065, USA

**Keywords:** hepatocellular carcinoma, biliary tract cancers, checkpoint inhibitor, drug combination, immunology, tumor microenvironment

## Abstract

Hepatocellular carcinoma (HCC) and biliary tract cancers (BTC) display a poor prognosis with 5-year overall survival rates around 15%, all stages taken together. These primary liver malignancies are often diagnosed at advanced stages where therapeutic options are limited. Recently, immune therapy has opened new opportunities in oncology. Based on their high programmed death-ligand 1 expression and tumor-infiltrating lymphocytes, HCC and BTC are theoretically good candidates for immune checkpoint blockade. However, clinical activity of single agent immunotherapy appears limited to a subset of patients, which is still ill-defined, and combinations are under investigation. In this review, we provide an overview of (i) the biological rationale for immunotherapies in HCC and BTC, (ii) the current state of their clinical development, and (iii) the predictive value of immune signatures for both clinical outcome and response to these therapies.

## 1. Introduction

Hepatocellular carcinoma (HCC) is the most frequent primary liver cancer (65,000 and 42,000 new cases/year in Europe and the United States, respectively) and the third leading cause of cancer death worldwide [1]. Most HCCs (80–90% of cases) develop on underlying chronic liver diseases, which are mainly related to chronic infections by hepatitis B virus (HBV) or hepatitis C virus (HCV), alcohol, or fatty liver disease [2,3,4,5,6].

Biliary tract cancers (BTCs) account for 10,000 and 12,000 new cases/year in Europe and the United States, respectively, ranking as the second primary liver malignancy, in terms of incidence, after HCC [7,8]. BTCs are yet rare tumors, with less than six new cases per 100,000 people each year. BTCs (adenocarcinoma in 90% of cases) are a heterogeneous group of tumors with different anatomical, biological, prognostic, and therapeutic features, gathering: (i) peripheral or intra-hepatic cholangiocarcinoma (iCCA); (ii) perihilar cholangiocarcinoma (pCCA), also known as Klatskin tumors; (iii) distal cholangiocarcinoma (dCCA), which are often grouped with pCCAs under the appellation extra-hepatic cholangiocarcinoma (eCCA); (iv) and gallbladder carcinoma [9,10]. BTC incidence, which is higher in Asia, is increasing, mainly for iCCA [10]. Similarly to HCC, the key risk factors associated with BTCs are cirrhosis, chronic infections by HBV and HCV, obesity, alcohol consumption, as well as chronic inflammatory biliary tract diseases such as sclerosing cholangitis and liver fluke infections in endemic areas (Asia) [10,11].

HCCs and BTCs are among the cancers with the poorest prognosis with a 5-year overall survival (OS) around 15%, all stages taken together. The only treatment with curative intent and offering the longest survivals for these primary liver malignancies is surgery (liver transplantation, surgical resection or radiofrequency ablation for HCCs; surgical resection for BTCs), but it is restricted to a minority of patients with localized diseases. Late diagnosis is one of the main reasons for the poor prognosis of these tumors. Sixty to 70% of BTC and 40% of HCC are diagnosed at advanced stages, where therapeutic options are limited. Indeed, many of the drugs tested in clinical trials in HCC and BTC patients yielded limited antitumor activity. Only a few of them were able to show partial efficacy such as angiogenesis inhibitors for HCC [12] and gemcitabine plus platinum chemotherapy for BTC [10]. Among the angiogenesis inhibitors tested in HCC, sorafenib and lenvatinib are approved for the first-line treatment of advanced HCC patients; and cabozantinib, regorafenib, and ramucirumab are therapeutic options in pre-treated patients [12].

Recently, immune therapies have become the main avenue in the treatment of solid tumors. These agents, mostly represented by anti-cytotoxic T lymphocyte antigen 4 (CTLA-4) and anti- programmed cell death 1 (PD-1/PD-L1) monoclonal antibodies (also known as immune checkpoint inhibitors or ICIs), have been studied in HCCs and BTCs. CTLA-4 and PD-1 are receptors expressed by T-cells that regulate immune responses at the priming phase in lymph nodes and at the effector phase in the tumor, respectively [13]. The restoration of the immune function of “exhausted” T cells and the depletion of immunosuppressive regulatory T lymphocytes (Treg) using monoclonal antibodies targeting these receptors brought about striking efficacy in several malignancies [13,14]. Approximately 20% of HCCs and BTCs are theoretically good candidates for ICIs based on their high level of expression of programmed death-ligand 1 (PD-L1) and tumor-infiltrating lymphocytes (TILs) [15,16]. However, clinical activity of ICI monotherapy appears restricted to a subset of patients, which are still ill-defined. Converting cold tumors into immunogenic tumors is one of the main therapeutic objectives of immuno-oncology research [17]. Several combinations are under investigation in HCC and BTC to this aim, with encouraging results. In this review, we provide an overview of (i) the biological rationale for immune therapies in HCC and BTC, (ii) the current state of their clinical development, and (iii) the predictive value of immune signatures for both clinical outcome and response to these therapies.

## 2. Hepatocellular Carcinoma

### 2.1. Biological Rationale for Immune Therapies

#### 2.1.1. Liver Is an Immunological Organ

The liver has an essential role in the immunological defense working as a biological sensor and active “filter” against infections of blood and digestive origin by secreting proteins and pro-inflammatory cytokines [18,19]. Anatomically, the liver is permanently exposed to exogenous antigens present in food and microbiota and contains many immune cells, from both innate and adaptive immune system. The fine-tuning of immune activation is crucial to avoid inadequate immune response against non-pathogenic molecules and self-antigens [18].

#### 2.1.2. Rationale for Using ICI in HCC

In patients who developed HCC in the context of drug-induced immunosuppression, spontaneous tumor regressions were observed following the discontinuation of immunosuppressants, restoring the ability of cytotoxic T-cells to identify and eliminate cancer cells [20]. These reports highlighted the dynamic relationship between the immune system and hepatocarcinogenesis.

PD-L1 ligand is detected (i) on tumor cells (threshold ≥ 1%) and (ii) on immune cells within the tumor microenvironment [21,22]. In a cohort of 217 resected HCCs, 75% of tumors expressed PD-L1, and PD-L1 expression was associated with aggressive tumor features (high alpha-fetoprotein (AFP) levels, satellite nodules, poor differentiation, vascular invasion) [21]. In addition, in a series of 956 HCCs, 25% of the samples displayed favorable immune microenvironments for ICIs with CD8+ TILs; high expression of CTLA-4, PD-1 and PD-L1; low Treg infiltration; and few tumor-associated macrophages (TAMs) [15]. Taken together, the observation of high PD-L1 expression and CD8 lymphocyte infiltration in a subset of HCC supports the use of ICIs.

#### 2.1.3. Rationale for Combining Angiogenesis Inhibitors and ICI in HCC

Vascularization of HCC tumors is rich and comes essentially from arterial blood vessels secondary to overexpression of pro-angiogenic growth, mainly, vascular endothelial growth factor A (VEGFA) and platelet-derived growth factor (PDGF) [23,24]. Amplifications of *VEGFA* gene (found in 4% to 8% of HCCs) induce both neoangiogenesis and tumor proliferation owing to the recruitment and activation of macrophages, which release hepatocyte growth factor (HGF) [25]. Besides, VEGFA is predominantly produced by tumor cells, TAMs, and cancer-associated fibroblasts (CAFs) [26]. VEGFA exerts immunomodulatory effects in many solid tumors [27]. Indeed, VEGFA drives the recruitment of VEGFR2-expressing Treg and decreases T cell extravasation at the tumor–endothelium interface [28]. VEGFA and pro-inflammatory cytokines induce a selective FasL expression at the surface of the tumoral endothelial cells, which allows the destruction of CD8+ T cells but not Treg [28], thereby acting as a barrier to antitumor T cell infiltration. Consequently, HCC are deprived in CD8+ T cells while immunosuppressive FoxP3+ Treg cells are abundant, resulting in an adverse immune cell imbalance. Therefore, vascular normalization using antiangiogenic agents has emerged as a new therapeutic strategy to modulate the immune microenvironment in HCC. VEGFA inhibition yielded reduced tumor growth together with an increase in the number of tumor-infiltrating CD8+ cells [29]. Similarly, translational studies in several solid tumors have shown that anti-VEGFA increases T cell infiltration into the tumors through vascular normalization, thereby enhancing the antitumor effect [30,31,32]. Thus, VEGFA plays a critical role in escaping antitumor immunity. The biological rationale for immune therapies in HCC is displayed in Figure 1.

### 2.2. Current State of Immune Therapies Clinical Development

#### 2.2.1. ICI Monotherapy

Two phase II trials evaluated tremelimumab (anti-CTLA-4) in HCC patients [33,34]. In the first study, 20 pre-treated patients with advanced HCC and chronic HCV infection received tremelimumab [34], resulting in an objective response rate (ORR) of 17.6% and a disease control rate (DCR) of 76.4%. The second study aimed at promoting the release of tumor antigens and increasing ICI efficacy in patients with advanced HCC through a combination with ablation therapy (chemoembolization or radiofrequency) [33]. Thirty-two patients were included but only 19 were radiologically evaluable because of early clinical progressions. An enrichment of intratumoral CD8+ T cells was observed 6 weeks after the start of the ICI in responders (26%). A decrease in circulating viral load of HCV was observed in both studies and no dose-limiting toxicities occurred. Nevertheless, the limited sample size prevented formal conclusions on the efficacy of CTLA-4 blockade in HCC, and these results should be taken with caution. Recently, the neo-adjuvant setting has been receiving increasing attention for emerging applications of anti-CTLA-4 monoclonal antibodies (NCT03682276).

Studies evaluating anti-PD-1/PD-L1 monoclonal antibodies as single agents in pre-treated patients with advanced HCC showed encouraging results (summarized in Table 1). The US FDA granted accelerated approval to pembrolizumab based on the results of KEYNOTE-224, a phase II trial in patients with advanced HCC in the second line setting. Nonetheless, the KEYNOTE-240 phase III trial, which compared pembrolizumab (anti-PD-1) vs. best supportive care (BSC) as second line therapy in 413 HCC patients, failed to achieve a statistically significant benefit on OS and PFS [35]. The depletion of the immune reserve in these pre-treated and progressive patients could be one of the reasons for this failure. The KEYNOTE-240 is ongoing in Asian patients (NCT03062358). Similarly, the Checkmate 459 phase III trial, which compared nivolumab to first-line standard sorafenib, also missed its primary endpoint (HR for OS: 0.85, *p*  =  0.0752) [36]. Nivolumab was recently approved for second-line treatment in the United States based on the phase I/II Checkmate 040 study in advanced HCC patients [37]. Currently, nivolumab is under evaluation in advanced HCC in combination with transforming growth factor beta (TGF-β) inhibitors (NCT02423343), indoleamine dioxygenase inhibitors (NCT03695250), intra-arterial therapies (chemoembolization, NCT03572582, and radioembolization, NCT03033446), as well as in the peri-operative setting (NCT03630640, NCT03383458). The combination of camrelizumab (anti-PD-1) with chemotherapy (FOLFOX4 or GEMOX) showed encouraging preliminary results in 34 treatment-naive HCC patients, with an ORR of 26.5% and a median progression-free-survival (PFS) of 5.5 months [38]. The synergy between ICI and cytotoxic agents remains to be further explored. The toxicity profile for anti-PD-1/PD-L1 in HCC was similar compared to other tumor types, whatever the molecule, and no cases of HBV reactivation have been detected.

#### 2.2.2. Combinations of ICIs and Other Immunotherapies

Past and ongoing combinations studies have aimed at evaluating anti-CTLA-4 and anti-PD-1/PD-L1 antibodies in advanced stage HCCs. The Checkmate-040 trial was a randomized multi-cohort phase II study assessing nivolumab plus ipilimumab in one of the cohorts at different dosing regimens in 148 pre-treated and naive patients [36]. Overall, the ORR was 31% including 5% (*n* = 7) of complete response. The highest ipilimumab dosing regimen (nivolumab 1 mg/kg + ipilimumab 3 mg/kg (4 doses) Q3W) displayed the most important benefit, with a DCR of 54% vs. 43% and 49% with other dosing regimens, and median OS of 23 months vs. 12 and 13 months with other regimens. Nonetheless, 37% of patients had a grade 3–4 ICI-related adverse event leading in 5% to discontinuation with this high-dose regimen. Similarly, the association of durvalumab and tremelimumab in a phase I study showed a DCR of 57.5% at 16 weeks with 20% of grade 3/4 adverse events leading in 7.5% to discontinuation [42]. Therefore, ICI combinations seem efficient but at the price of an increased toxicity and cost, requiring the implementation of biomarker studies in larger randomized clinical trials to guide patient selection (NCT02519348, NCT03298451, NCT01658878). ICI combinations are also currently evaluated in the neoadjuvant (NCT03510871, NCT03222076) and adjuvant (NCT03203304, NCT03638141) settings.

Furthermore, other modalities of immunotherapy are under evaluation in HCC, such as peptide and dendritic cell-based vaccines, oncolytic viruses and chimeric antigen receptor (CAR) T-cell therapies. Peptide vaccines based on HCC-associated antigens (AFP, glypican-3, telomerase reverse transcriptase) have shown limited efficacy although inducing T-cell-specific responses [47]. Dendritic cell-based vaccines have been evaluated in many small studies in HCC with encouraging results providing hope for further development [48]. Besides, early-phase trials evaluating adenovirus and vaccinia virus in HCC patients, which selectively target and replicate in cancer cells, have reported objective tumor responses [49]. A randomized phase III study that is still ongoing is evaluating the combination of vaccinia virus with sorafenib (NCT02562755). Moreover, clinical trials of CAR-T-cells for solid tumors have been recently launched after proving efficacy in hematologic diseases [50]. Their clinical development in HCC and hepatoblastoma followed promising results from pre-clinical studies and mostly involves AFP, glypican-3, epithelial growth factor receptor, and HBV antigens (NCT03618381; NCT04093648) [51]. However, preliminary trials in HCC showed modest anti-tumor responses [51,52]. Strategies such as identifying more immunogenic antigens, improving T cells tumor penetration, and combining CAR-T cells with other antitumor drugs are explored to improve the efficacy of these immune strategies in HCC and reduce the risk of fatal adverse events (e.g., cytokine release syndrome) [51]. Finally, combining ICI with cellular therapies or other immunotherapeutic approaches (i.e., cancer vaccines and oncolytic viruses) might be synergistic and improve OS of HCC patients (NCT03071094) [53].

#### 2.2.3. Combinations of Angiogenesis Inhibitors and ICI

The “Barcelona Clinic Liver Cancer” (BCLC) classification taking into account the patient performance status (PS) (European Collaborative Oncology Group [ECOG] score), the underlying liver function (Child Pugh score), and HCC tumor burden, is recommended to choose the best treatment for HCC patients [12,24,54,55,56]. For advanced HCC patients with ECOG PS 0–2 and a preserved liver function, angiogenesis inhibitors are the only recommended treatment. Sorafenib and lenvatinib can be used in first line, while cabozantinib, ramucirumab and regorafenib have become new therapeutic options for second line and beyond [54].

Patients with unresectable HCC are currently included in trials exploring combinations of antiangiogenics and ICI. First, the phase III IMbrave150 trial evaluating bevacizumab and atezolizumab in first line vs. sorafenib showed improvement in PFS (median 6.8 vs. 4.3 months, *p* < 0.0001) and OS (median not reached vs. 13.2 months, *p* = 0.0006) in 501 patients. ORR was increased with the combination therapy (27% vs. 12%, *p* < 0.0001) [43]. Grade 3–4 toxicities (mostly hypertension, abnormal liver tests and autoimmune manifestations) were observed in half of patients. As previously reported with ICI monotherapy, patients with chronic HCV infection (43%) responded better, as well as those with AFP ≥ 400 ng/mL. Notably, after a median follow-up of 7.2 months, 83% of responses were maintained. The results of this positive study are a breakthrough, setting the combination of ICI and angiogenesis inhibitors as a future standard of care in the management of HCC, although the high cost may limit access to these drugs. Secondly, the preliminary results of the early-phase study evaluating the association between lenvatinib and pembrolizumab showed an ORR of 44.8% in the 67 evaluable patients with acceptable toxicity [44]. These encouraging results led to the initiation of a phase III study evaluating lenvatinib plus pembrolizumab as first line therapy for patients with advanced HCC [57]. Thirdly, a communication about a phase Ib combining axitinib and avelumab in 22 naive HCC patients reported an ORR of 31.8% according to modified response evaluation criteria in solid tumors (RECIST), with an acceptable safety profile [46]. Finally, nivolumab is currently evaluated in combination with bevacizumab (NCT03382886), lenvatinib (NCT03418922), cabozantinib (NCT03299946), and ipilimumab and cabozantinib in a cohort study within the Checkmate 040 trial [37]. The association of durvalumab and bevacizumab is also explored in an ongoing randomized phase III trial (NCT03778957). Ongoing combination trials evaluating immune therapies in HCC are displayed in Table 2.

Overall, the combination of anti-angiogenics and ICIs appears to be synergic. Although the ORR results are encouraging, toxicity and survival results must be confirmed by larger clinical trials. Moreover, further biomarker studies are mandatory to determine which patients would benefit the most from such strategies.

Recent research studies have created classifications of the HCC immune microenvironment based on transcriptomic and immune pathways analysis. First, a pan-tumor study revealed six immune patterns among HCC samples [57]. Interestingly, cluster 1 (wound healing profile, 10%) was associated with a higher expression of angiogenic genes, supporting the idea that antiangiogenics may be useful for these patients. While cluster 3 (inflammatory, 30%) had the best survival, the dominant subset was cluster 4 (depleted in lymphocytes, 40%) but had no prognostic effect. Clusters 5 (immunologically calm) and 6 (TGF-β dominant) were uncommon (<5%) in HCC. Significantly, CD8+ T cells number was positively associated with the predicted neoantigen quantity, and clusters 2 and 3 were enriched in neoantigens, which may account for their higher CD8/Treg ratios compared to other clusters. Secondly, a study conducted by the The Cancer Genome Atlas (TCGA) consortium in 196 HCC patients highlighted that 22% of HCCs had significant or moderate lymphocyte infiltration, while 25% were poorly infiltrated [58]. Importantly, lymphocyte-rich tumors strongly expressed interferon-γ (IFN-γ) and the immune checkpoints CTLA-4, PD-1 and PD-L1. The authors also showed that virally-induced tumors (HBV and HCV-related) had the same immune profiles as non-virus-related tumors. Finally, in the above-mentioned cohort of 956 HCC samples, half of the 25% lymphocytes-rich tumors displayed an IFN-γ pathway signature [15] that has been reported to be predictive of ICI efficacy in other solid tumors [59]. The other half was characterized by exhausted immune responses and a more aggressive phenotype mainly regulated by the TGF-β pathway. The combination of angiogenesis and TGF-β inhibitors might be relevant to explore in this specific patient population. Finally, cold HCC tumors with low response to ICIs were enriched in WNT-β catenin pathway alterations (*CTNNB1* and *AXIN1* mutations) [15,60,61,62], supporting the idea that tumor molecular alterations may shape the immune microenvironment.

Overall, HCC is composed of several immune phenotypes with different survival and response to therapeutics raising the question of personalized therapeutics guided by microenvironment profiles. Nevertheless, it is warranted to continue research about immune cells interactions in HCC to improve therapeutic strategies and to bring such clinical approaches into practice.

## 3. Biliary Tract Cancers

Most patients with BTCs (60%) are diagnosed at an advanced stage and are not accessible to curative-intent surgery. Therapeutic options in this setting are limited and are based on systemic chemotherapy (first line with gemcitabine plus cisplatin, second line with 5-fluorouracil plus oxaliplatin) combined with supportive care to improve patient OS and health-related quality of life [10,63]. Chemotherapy provides only limited survival benefit in advanced BTC and the development of new therapeutic approaches with companion biomarkers to predict response is crucial for improving patient outcomes [64].

### 3.1. Biological Rationale for Targeted Therapies and Immune Therapies

BTC shares the same tumor niche as HCC. Some BTCs are liver-predominant diseases, indeed almost 50% of iCCA are located only in the liver and have a better OS compared to other BTCs [63]. BTC displays a prominent stromal component promoting tumor progression and resistance to therapy. The stroma of BTCs is a dynamic environment consisting of CAFs, immune cells, and endothelial cells surrounding tumor cells and interacting together. The microenvironmental conditions and barriers have to be taken into account for immunotherapies to succeed in BTCs.

#### 3.1.1. Molecular Alterations

BTCs are a group of tumors with different clinical and molecular characteristics. In recent years, the development of high-throughput genomic and transcriptomic analyses has revealed BTC molecular heterogeneity [65]. The complex landscape of BTC genetic alterations has been deciphered, including some druggable mutations or rearrangements, paving the way for personalized targeted therapies [66,67,68]. Interestingly, some alterations were enriched in specific primary tumor locations [69] (Figure 2). Hence, isocitrate dehydrogenase 1/2 (*IDH1/2*) mutations and fibroblast growth factor receptor 2 (*FGFR2*) gene fusions were each described in 10–20% of iCCA [70]. IDH and FGFR alterations represent the two main “modern” therapeutic targets in BTCs and are the most advanced in their clinical development. [71,72].

#### 3.1.2. Angiogenesis

Contrary to eCCAs, iCCAs are well-vascularized tumors and display arterial phase contrast-enhancement on imaging, similar to HCC [68]. In addition, iCCAs display a higher expression level of VEGFA and microvessel density compared to eCCAs [73,74]. Taken together, these data gave a rationale for treatment with antiangiogenics in this specific subgroup of BTCs. While most antiangiogenic agents in unselected BTCs failed to show an improvement in patient survival, in some small phase II studies restricted to patients with iCCA, sunitinib monotherapy [75] or bevacizumab in combination with FOLFIRI [76] in second-line yielded a median OS reaching 9.6 and 20 months, respectively. Therefore, antiangiogenics may be of interest specifically in patients with iCCA, and they are under exploration in ongoing studies (anlotinib [NCT03940378], apatinib [NCT03521219]). Noticeably, recent phase II trials suggested activity of regorafenib in patients with pre-treated BTC regardless of the primary tumor location [77,78].

#### 3.1.3. Immune Microenvironment

Knowledge about the BTC immune microenvironment remains scarce and needs to be further characterized [79]. Immunohistochemistry studies showed an association between CD8-positive TILs, natural killer lymphocytes, and major histocompatibility complex class I expression and prolonged survival; whereas M2-macrophage and neutrophil infiltrates were associated with early recurrence and death; and T regulatory cells (FOXP3-positive) showed inconsistent prognostic value [16,80,81,82,83,84]. Although poorly represented in the TCGA initiative [85], BTCs seems to be rich in immune cells in 70% of cases and depleted in lymphocytes in 30%. Immune-inflamed BTC shows a predominantly balanced macrophage-to-lymphocyte ratio with high expression of immune checkpoints. PD-L1 expression by tumor cells has been observed in 9% to 30% and associated with a higher density of TILs [16]. PD-L1 expression and the presence of TILs have been associated with a response to ICIs [86].

#### 3.1.4. Subgroups Sensitive to Immune Therapies

Molecular studies of BTCs identified a subgroup of tumors that may be good candidates for ICIs. Nakamura et al. [66] described four subtypes of BTCs according to hierarchical clustering of global gene expression levels of 260 tumors. All patients with *IDH1* mutation or *FGFR2* fusion were associated with Cluster 3, including only iCCA. On the other hand, almost 40% of patients were classified in Cluster 4 and associated with a poor prognosis. This subgroup showed a higher expression of PD-L1 and eight other immune checkpoint genes (*LAG3*, *CTLA4*, *PDCD1*, *TNFRSF9*, *BTLA*, *IDO1*, *HAVCR2*, and *TNFRSF4*), suggesting an immune environment favorable to ICIs (Figure 2). Moreover, hypermutated cases, where a high mutation load was expected to create abundant tumor-specific neoantigens and thereby immunogenic tumors, were also significantly enriched in Cluster 4 [67]. Other classifications compared the profile of fluke-related (*Opisthorchis viverrini* and *Clonorchis sinensis*, mainly found in Asia) to fluke-negative BTCs [67,87]. With an integrative clustering, Jusakul et al. [67] also defined four molecular subtypes of BTCs. Clusters 3 and 4 included intra-hepatic tumors which were mostly fluke-negative. While *IDH1/2* mutations and *FGFR* alterations were enriched in Cluster 4, Cluster 3 exhibited specific upregulation of immune checkpoint genes (*PD-1*, *PD-L2*, and *BTLA*) [67]. Thereby, immunogenic iCCA seems mutually exclusive with *IDH*/*FGFR* driven iCCA.

In addition, a subset of BTCs (about 10% of iCCAs and dCCAs and 5% of pCCAs and gallbladder carcinomas) displays DNA mismatch repair deficiency (dMMR) and/or microsatellite instability (MSI; Figure 2) [88,89]. dMMR/MSI phenotype is characterized by a high load of neoantigens that activate antitumor T-cell response and has been associated with durable responses to ICIs in various tumor types. Similar to other MSI tumors, MSI BTCs are sensitive to anti-PD1 therapy, with median OS reaching 24 months [88,90].

The identification of BTC subtypes with specific genomic alterations, some of which may be druggable, led to the hypothesis that treatment may be personalized for each patient based on molecular profiling of tumors [65]. However, although the first success emerged with IDH and FGFR targeting [71,72], there is currently no treatment development in selected subpopulations associated with an immune signature.

### 3.2. Current State of Immune Therapies Clinical Development

Several strategies have been explored to modulate anti-tumor immunity of the host in BTCs, using vaccines, adoptive cell therapies, or ICIs [13,91].

#### 3.2.1. ICI Monotherapy

Early studies of ICIs in BTCs, regardless of MSI status, showed the first signal of activity. Pembrolizumab monotherapy has been tested in a phase Ib study in pre-treated patients with PD-L1-positive (>1% positive cells) BTC [92]. Thirty-seven (42%) patients out of 89 screened had PD-L1-positive tumors, and 24 of them received pembrolizumab. The ORR was 17%, including five long-lasting responses (>40 weeks). The safety profile was manageable, with low rates of immune-mediated toxicity [92]. In a phase II study in 61 patients with pre-treated BTCs with a PD-L1 combined positive score (CPS, number of PD-L1 staining cells [tumor cells, lymphocytes, macrophages] divided by the total number of viable tumor cells, multiplied by 100) ≥1, pembrolizumab monotherapy yielded ORR of 6.6%, median PFS of 2.0 months, and a favorable toxicity profile. Prolonged tumor responses were observed in some patients. In this same study, 34 patients with CPS < 1 received the same treatment, however, tumor control was lower: ORR was only 2.9% and median PFS was 1.9 months [93]. Of note, none of the patients across the two studies were MSI [92]. Interestingly, two cases of complete disease resolution were reported after pembrolizumab alone in second-line treatment [94].

Nivolumab showed similar results in a phase II study in 54 chemo-refractory BTC patients. The median PFS was 4.0 months, the DCR was 60%, and 10 (22%) patients had a partial response. All patients who responded were microsatellite stable [95]. All of these studies reported a favorable safety profile, with 13–20% of grade 3–5 treatment-related adverse events and 15–20% of immune-mediated toxicity [92,93,95]. Overall, ICI monotherapy showed encouraging results, especially in PD-L1 selected population, regardless of the MSI status. Data from phase III studies are lacking and there is no validated strategy to date.

#### 3.2.2. ICI in Combination and Other Immunotherapies

A phase I study evaluated durvalumab alone (*n* = 42) or combined with tremelimumab (*n* = 65) in Asian patients. It showed limited ORR (4.7% and 7.7% with monotherapy and combination with tremelimumab, respectively) but promising DCR at 12 weeks (6.7% and 32.2%, respectively), median duration of response (9.7 and 8.5 months, respectively), and median OS (8.1 and 10.1 months, respectively) in pre-treated BTC patients. The grade ≥3 treatment-related adverse events were similar in these two cohorts (19% and 23%, respectively) [96]. Nivolumab was also tested alone or in combination with cisplatin plus gemcitabine in a phase I study in Japanese patients. Median OS and PFS were 15.4 and 4.2 months, respectively, with the combined therapy. Of note, 11 of 30 patients had an objective response [97].

Several trials are ongoing evaluating anti-PD-1/PD-L1 antibodies as monotherapy or in combination; they are summarized in Table 3. A bifunctional anti-PD-L1/TGFβ trap fusion protein (M7824) has also shown encouraging activity as monotherapy, with long-term responses in eight (27%) out of 30 Asian patients with pre-treated BTC [98]. Phase II-III trials with M7824 are ongoing (NCT03833661; NCT04066491).

Vaccines [99,100,101] and cellular therapies [102,103], as CAR-engineered T cell targeting erb-b2 receptor tyrosine kinase 2 (ERBB2, commonly referred to as HER2) [104], have also been explored in early studies (NCT03801083; NCT03042182). Moreover, a growing interest has developed regarding tumor stroma as well as its role in supporting and promoting tumor growth and its involvement in resistance to treatment. By analogy with pancreatic ductal adenocarcinoma, which is also characterized by an abundant desmoplastic stroma, strategies targeting tumor microenvironment (e.g., CAFs and other components the microenvironment) are emerging in BTCs [68,69,105].

Overall, immunotherapies are under clinical development in BTCs. An important issue to improve the activity of ICIs is the identification of biomarkers to guide patient selection for these strategies.

### 3.3. Predictive Value of Immune Signatures

The identification of prognostic and predictive biomarkers to guide patient selection for therapy has become a major issue in BTC in recent years. Unfortunately, PD-L1 expression has been inconsistently associated with sensitivity to ICIs in various tumors. Owing to the activity of ICIs in dMMR/MSI tumors [88], the United States Food and Drug Administration has granted a tumor-independent, agnostic approval for pembrolizumab in pre-treated patients with advanced dMMR/MSI solid tumors, including BTCs. However, dMMR/MSI is detected in only 5–10% of BTCs. Thus, the population of BTC patients who are responders to ICIs (up to 20%) extends beyond MSI-positive tumors [93].

Beyond MSI, hypermutated cancers have been associated with durable clinical benefit from ICIs and may be excellent candidates for immunotherapy. Wardell et al. [106] analyzed the genomic features of 412 BTC samples from Japanese and Italian patients. Eleven percent of BTC patients harbored deleterious germline mutations of cancer-predisposing genes such as *BRCA1*, *BRCA2*, *RAD51D*, *MLH1*, *MSH2*, *POLD1*, *POLE*, *TP53*, and *ATM*. Interestingly, all of the hypermutated samples display somatic or germline mutations in at least one known DNA mismatch repair gene (e.g., *MLH1, MSH2*) or DNA polymerase (e.g., *POLD1, POLE*) [106]. Of note, tumor mutational burden (≥6 mutations/Mb) was significantly higher in eCCA and gallbladder carcinoma (18% and 22%, respectively) versus iCCA (13%) [107].

As described above, Thorsson et al. [85] proposed six common immune patterns that may predict tumor response to ICIs. However, no predictive immune signature of response or resistance has been identified so far in BTCs. Biomarkers to predict the response to immune therapy in BTC still remain to be defined, highlighting the importance of translational studies on tumor and blood samples in ongoing trials [108,109].

## 4. Conclusions

HCCs and BTCs share a common unique microenvironment and immune niche (i.e., the liver). 

HCCs are immunogenic but immunosuppressed and highly vascularized tumors. ICI monotherapies have been disappointing, but recent results of combinations with antiangiogenics are very encouraging.

BTCs are an anatomically, molecularly and therapeutically heterogeneous group of tumors, with a poor prognosis. Immunotherapies are under clinical development in BTC and show activity in an ill-defined subgroup of immunogenic tumors, extending beyond MSI phenotype.

Most promising results come from approaches combining targeted therapy on one side and immune therapy to restore or boost the function of immune cells on another side, not only in liver cancers but also in other tumor types. For example, in advanced gastric or colorectal cancers, the combination of regorafenib with nivolumab has been tested with interesting activity, not limited to MSI patients [109]. Such combinations would warrant evaluation in HCC and BTC. However, they are associated with increased toxicity; optimization of the handling and schedule of these agents (e.g., dose adjustment, sequential administration) may improve their tolerance.

In both HCCs and BTCs, only a subset of patients benefits from ICIs. Classical biomarkers (e.g., mutation burden, PD-L1 expression, TILs) display variable sensitivity and specificity across tumor types. Response to ICIs may be modulated by the microenvironment immune pattern, which is currently extensively explored as a source of predictive biomarkers. Intratumoral heterogeneity is a challenge for this type of biomarker, particularly in the advanced setting, where limited quantities of tumor material (biopsies) are available. Liquid biopsies (e.g., circulating DNA, RNA, vesicles, and tumor cells) may provide useful information but are so far mainly prognostic rather than predictive markers, and are not validated yet for routine practice. Overall, the process of companion biomarker identification is a continuous, bench-to-bedside and bedside-to-bench process.

Future progress will come from clinical trials evaluating rationale combinations and will include extensive translational research on blood and tumor to better understand the mechanisms of response or resistance to immune therapy and guide patient selection for these approaches.

## Figures and Tables

**Figure 1 cancers-12-00077-f001:**
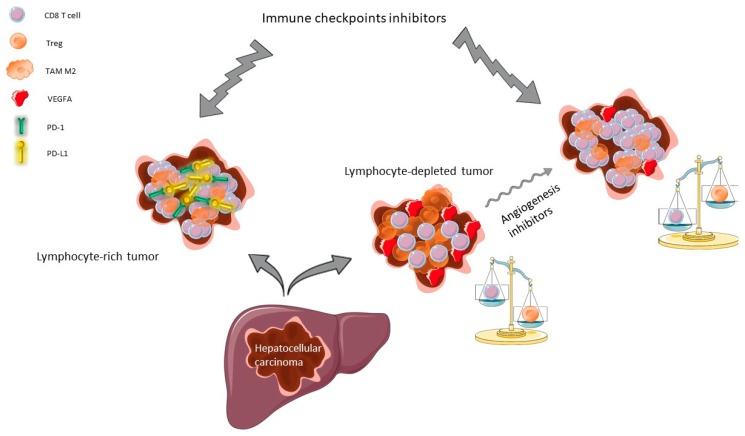
Biological rationale for use of immune checkpoints inhibitors (ICI) in hepatocellular carcinoma (HCC). Approximately 25% of HCC are highly infiltrated in cytotoxic lymphocytes (CD8 T cells) with a strong expression of the immune checkpoints such as programmed death-1 (PD-1) and its ligand (PD-L1) (**left panel**), whereas about 40% of HCC are depleted in cytotoxic lymphocyte because of the immunosuppressive effect of the vascular endothelial growth factor A (VEGFA) (**right panel**). Inhibition of angiogenesis reverses the CD8 T cell/T regulatory cell (Treg) ratio, enabling ICI efficacy in HCC. TAM: tumor-associated macrophage.

**Figure 2 cancers-12-00077-f002:**
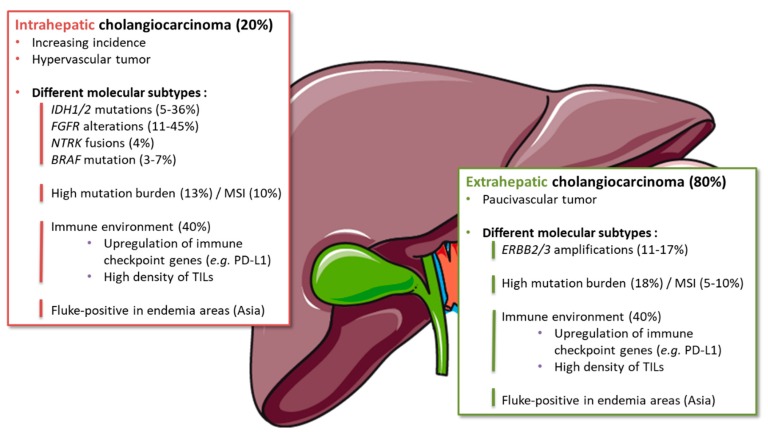
Molecular and immune subgroups in the different anatomical subtypes of biliary tract cancer. Abbreviations: ERBB: erb-b2 receptor tyrosine kinase; FGFR: fibroblast growth factor receptor; IDH: isocitrate dehydrogenase; MSI: microsatellite instability; NTRK: neurotrophic tyrosine receptor kinase; PD-L1: programmed death ligand-1; TIL: tumor infiltrating lymphocytes.

**Table 1 cancers-12-00077-t001:** Summary of clinical trials of immune therapies (single agent or in combination with angiogenesis inhibitors) in patients with advanced hepatocellular carcinoma (HCC) ^1^.

Type of Immunotherapy	Molecules	Trial	Phase	*n*	Population	mOS	mPFS	ORR	DCR
Anti-CTLA-4	Tremelimumab	Sangro et al. [34]	II	20	Pre-treated	8.2 mo	6.5 mo	17.6%	76.4%
		Duffy et al. [33]	II	32	Pre-treated	12.3 mo	7.4 mo	26.3%	63%
Ant-PD1	Pembrolizumab	Finn et al. [35]	III	413	Pre-treated	12.9 mo	4.9 mo	16.9%	NA
	Nivolumab	Yau et al. [37]	III	371	Naive	16.4 mo	3.7 mo	15%, 4% CR	NA
		El-Khoueiry et al. [37]	I/II	262	Pre-treated and naive	NR	4 mo	20%, 1% CR	64%
	Cemiplimab	Pishvaian et al. [39]	I	26	Pre-treated	NR	3.7 mo	19.2%	73%
Anti-PD-L1	Durvalumab	Wainberg et al. [40]	I/II	39	Pre-treated	13.2 mo	NA	10.3%	33% at 6 m
Anti-PD-1 + Anti CTLA-4	Nivolumab + ipilumumab	Yau et al. [41]	II	148	Pre-treated	NR	NA	31%, 5% CR	49%
	Durvalumab + tremelimumab	Kelley et al. [42]	I	40	Pre-treated and naive	NA	NA	15%	57.5% at 4 m
Angiogenesis and immune checkpoints inhibitors	Atezolizumab + bevacizumab	Cheng et al. [43]	III	501	Naive	NR	6.8 mo	37%	NA
	Pembrolizumab + lenvatinib	Llovet et al. [44]	Ib	67	Naive	NA	NA	45%	91%
	Camrelizumab + apatinib	Xu et al. [45]	Ib	16	Pre-treated	NR	5.8 mo	50%	93.8%
	Avelumab + axitinib	Kudo et al. [46]	I	22	Naive	NR	5.5 mo	31.8%	NA
Cytotoxic agents and Anti-PD-1	FOLFOX4 or GEMOX + camrelizumab	Qin et al. [38]	Ib	34	Naive	NR	5.5 mo	26.5%	79.4%

^1^ CR: complete response; CTLA-4: Cytotoxic T lymphocyte-associated protein 4; DCR: disease control rate; mo: months; mOS: median overall survival; mPFS: median progression-free-survival; *n*: number of randomized patients; NR: not reached; NA: not available; ORR: objective response rate; PD-1: programmed cell death-1; PD-L1: Programmed death-ligand 1.

**Table 2 cancers-12-00077-t002:** Ongoing combination trials evaluating immune therapies in hepatocellular carcinoma ^1^.

Molecule	Targets	Phase	Population	ClinicalTrial.gov Reference
**Immunotherapies in Combotherapy**				
Nivolumab plus ipilimumab	PD-1 and CTLA-4	I-III	Localized HCC Intermediate stage HCC Advanced HCC, L2 Advanced HCC, L1	NCT03510871 NCT03222076 NCT03203304 NCT01658878 NCT04039607
Durvalumab plus tremelimumab	PD-L1 and CTLA-4	III	Advanced HCC, L2	NCT03298451
LY3321367 plus LY3300054	PD-L1 and TIM-3	I	Advanced HCC, L2	NCT03099109
**Immunotherapies in association with angiogenesis inhibitors**				
Nivolumab plus bevacizumab	PD-1	I	Advanced HCC, L2	NCT03382886
Nivolumab plus lenvatinib	PD-1	I-II	Advanced HCC, ≥ L1	NCT03418922 NCT03841201
Nivolumab plus sorafenib	PD-1	I-III	Advanced HCC, L1	NCT02576509 NCT01658878 NCT03439891
Nivolumab plus cabozantinib	PD-1	I-II	Locally advanced HCC Advanced HCC, L1	NCT03299946 NCT01658878
Nivolumab plus GT90001	PD-1	I/II	Advanced HCC	NCT03893695
Pembrolizumab plus lenvatinib	PD-1	III	Advanced HCC, L1	NCT03713593
Pembrolizumab plus sorafenib	PD-1	I/II	Advanced HCC, L1	NCT03211416
Pembrolizumab plus regorafenib	PD-1	I	Advanced HCC, L1	NCT03347292
Atezolizumab plus cabozantinib	PD-L1	III	Advanced HCC, L1	NCT03755791
Atezolizumab plus bevacizumab	PD-L1	III	Advanced HCC, L1	NCT03434379
Avelumab plus axitinib	PD-L1	I	Advanced HCC, L1	NCT03289533
Durvalumab plus bevacizumab	PD-L1	III	Localized HCC Locally advanced HCC	NCT03847428 NCT03778957
Durvalumab plus tivozanib	PD-L1	I/II	Advanced HCC, L1	NCT03970616
Camrelizumab plus apatinib	PD-1	I III	Advanced HCC, L2 Advanced HCC, L1	NCT02942329 NCT03764293
Spartalizumab plus sorafenib	PD-1	I	Advanced HCC, L1	NCT02988440
Tislelizumab plus sorafenib	PD-1	III	Advanced HCC, L1	NCT03412773
Sintilimab plus IBI305	PD-1	III	Advanced HCC, L1	NCT03794440
**Immunotherapies in association with loco-regional therapies (radiofrequency ablation, radiotherapy, or intra-arterial treatments)**				
Durvalumab plus tremelimumab	PD-L1 and CTLA-4	II	Intermediate stage HCC	NCT03638141
Nivolumab ± ipilimumab	PD-1	I–II	Advanced HCC Intermediate stage HCC	NCT03033446 NCT03572582 NCT03143270 NCT02837029 NCT03203304 NCT03812562 NCT03630640
Pembrolizumab	PD-1	I–II	Locally advanced	NCT03099564 NCT03397654 NCT03867084 NCT03316872
**Immunotherapies in association with another therapy**				
Camrelizumab plus chemotherapy	PD-1	I	Advanced HCC, L1	NCT03092895
Nivolumab plus TGF-β inhibitor	PD-1	I/II	Advanced HCC, L2	NCT02423343
Nivolumab plus indoleamine dioxygenase inhibitor	PD-1	I/II	Advanced HCC, L1	NCT03695250
Nivolumab plus cereblon modulator	PD-1	I/II	Advanced HCC, L2	NCT02859324
Nivolumab plus invariant Natural Killer T cell agonist	PD-1	I/II	Advanced HCC, ≥L2	NCT03897543
Nivolumab plus Pexa-Vec oncolytic immunotherapy	PD-1	I/II	Advanced HCC, L1	NCT03071094
Pembrolizumab plus monoclonal antibody against phosphatidylserine	PD-1	II	Advanced HCC, L1	NCT03519997
Spartalizumab plus FGFR4 inhibitor	PD-1	I/II	Advanced HCC, L2	NCT02325739
Spartalizumab plus MET inhibitor	PD-1	I/II	Advanced HCC, L2	NCT02795429

^1^ Abbreviations: CTLA-4: cytotoxic T-lymphocyte–associated antigen 4; FGFR: fibroblast growth factor receptor; HCC: hepatocellular carcinoma; L1: first line treatment; L2: second line treatment; PD-1: programmed death 1; PD-L1: programmed death ligand-1; TGFβ: transforming growth factor beta; TIM-3: T cell immunoglobulin and mucin domain 3.2.3. Predictive Value of Immune Signatures.

**Table 3 cancers-12-00077-t003:** Ongoing phase II or III trials evaluating immune therapies in biliary tract cancer ^1^.

Molecule	Targets	Phase	Population	ClinicalTrial.gov Reference
**Immunotherapies in monotherapy or combotherapy**				
Nivolumab plus ipilimumab	PD-1 and CTLA-4	IIR	Advanced CCA, ≥L1	NCT03101566 NCT02834013
Pembrolizumab	PD-1	II	Advanced CCA, L2	NCT03110328 NCT02628067
Nivolumab	PD-1	II	Advanced CCA, ≥L2	NCT02829918
STI-3031	PD-L1	II	Advanced CCA, ≥L2	NCT03999658
**Immunotherapies in association with chemotherapy**				
Durvalumab	PD-L1	III	Advanced CCA, L1	NCT03875235
KN035	PD-L1	III	Advanced CCA, L1	NCT03478488
Pembrolizumab	PD-1	II-III	Advanced CCA, ≥L1	NCT03260712 NCT03111732 NCT04003636
Durvalumab plus tremelimumab	PD-L1 and CTLA-4	IIR	Advanced CCA, ≥L1	NCT03473574 NCT03046862 NCT03704480
SHR-1210	PD-1	II	Advanced CCA, ≥L1	NCT03486678
Toripalimab	PD-1	II	Advanced CCA, L1	NCT03796429 NCT03982680 NCT04027764
**Immunotherapies in association with radiation or ablative therapies**				
Nivolumab ± ipilimumab	PD-1 and CTLA-4	IIR	Advanced CCA, ≥L2	NCT02866383
Durvalumab plus tremelimumab	PD-L1 and CTLA-4	II	Advanced CCA, ≥L2	NCT03482102 NCT02821754
Camrelizumab	PD-1	II	Advanced CCA, L1	NCT03898895
**Immunotherapies in association with another therapy**				
SHR-1210 plus apatinib	PD-1	IIR	Advanced CCA, ≥L2	NCT03092895
Atezolizumab plus cobimetininb (MEK inhibitor)	PD-L1	IIR	Metastatic CCA, ≥L2	NCT03201458
Pembrolizumab plus lenvatinib	PD-1	II	Advanced CCA, ≥L2	NCT03797326 NCT03895970
Toripalimab plus axitinib	PD-1	II	Metastatic CCA, L2	NCT04010071
Durvalumab plus olaparib	PD-L1	II	Advanced CCA, *IDH1/2* gene mutation, ≥L2	NCT03991832
Nivolumab plus rucaparib	PD-1	II	Advanced CCA, ≥L2	NCT03639935
Atezolizumab plus DKN-01 (DKK1 inhibitor)	PD-L1	II	Advanced CCA, ≥L2	NCT03818997
Nivolumab plus DKN-01 (DKK1 inhibitor)	PD-1	II	Advanced CCA, ≥L2	NCT04057365
Pembrolizumab plus sargramostim (GM-CSF)	PD-1	II	Advanced CCA, ≥L1	NCT02703714
Nivolumab plus entinostat (HDAC inhibitor)	PD-1	II	Advanced CCA, ≥L2	NCT03250273
JS001 plus lenvatinib plus GEMOX	PD-1	II	Advanced iCCA, L1	NCT03951597

^1^ Abbreviations: IIR: phase II, randomized; CCA: cholangiocarcinoma; CSF1R: colony stimulating factor-1 receptor; CTLA-4: cytotoxic T-lymphocyte–associated antigen *4*; DKK1: dickkopf WNT signalling pathway inhibitor 1; GEMOX: gemcitabine plus oxaliplatin; GM-CSF: granulocyte-macrophage colony stimulating factor; HDAC: histone deacetylase; L1: first line treatment; L2: second line treatment; PD-1: programmed death 1; PD-L1: programmed death ligand-1.

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
