# Peer review of "Immune Therapy for Liver Cancers"

_cancers, 2019, doi:10.3390/cancers12010077_

Round 1

Reviewer 1 Report

In the current review the authors discuss the rationale of testing immunotherapy in liver cancer (biliary tract cancers and hepatocellular carcinoma) and summarize latest data on checkpoint blockers in these tumor entities.

Introduction:

The authors should briefly mention currently available systemic treatment options in HCC. The authors state that liver tumors highly express PD-L1 and tumor infiltrating lymphocytes. Please provide a reference. Notably, in the Checkmate459 phase III trial of nivolumab vs sorafenib in HCC, the percentage of patients with PD-L1 expression >1% was only 19%, which I would not consider high.

ICI monotherapy:

The authors should mention the phase II Checkmate 040 study, based on which nivolumab was approved for second-line treatment in the US.

Combination of ICIs:

The authors should mention that the Checkmate 040 was a multi-cohort study, and that the combination of nivolumab plus ipilimumab was tested in one cohort of these cohorts.

Combinations of angiogenesis inhibitors and ICIs:

45: an update was presented at ESMO 2019 including 67 patients. Please update. The authors should discuss the impact of the positive IMbrave150 study on the future management of HCC.

Others:

The authors should include a Table showing ongoing phase I-III combination trials in HCC, categorized according to combination partner (e.g., ICI, anti-angiogenic agent, anti-TGF-b, etc.) Some typos/grammar errors need to be corrected throughout the paper. The authors should include a paragraph on ‘Future perspectives’

Author Response

Paris, 22nd December 2019

Dear Editor,

We would like to thank you and the Reviewers for their very helpful comments and suggestions.

We attach a revised version of our manuscript cancers-675416 with tracked-in-red changes in the body of the text. Please find below the point-by-point responses to the Reviewer’s comments.

Reviewer #1:

In the current review the authors discuss the rationale of testing immunotherapy in liver cancer (biliary tract cancers and hepatocellular carcinoma) and summarize latest data on checkpoint blockers in these tumor entities.

Introduction:

The authors should briefly mention currently available systemic treatment options in HCC.

Thank you for your suggestion. We added the following text:

« Among the angiogenesis inhibitors tested in HCC, sorafenib and lenvatinib are approved for the first-line treatment of advanced HCC patients, and cabozantinib, regorafenib, and ramucirumab are therapeutic options in pre-treated patients [12]. »

The authors state that liver tumors highly express PD-L1 and tumor infiltrating lymphocytes. Please provide a reference. Notably, in the Checkmate459 phase III trial of nivolumab vs sorafenib in HCC, the percentage of patients with PD-L1 expression >1% was only 19%, which I would not consider high.

Thank you for your comment. PD-L1 expression in HCC and BTCs is around 20%, which is relatively high compared to other types of tumor. Expression of PD-L1 in immunogenic cancers such as lung cancer (PMID: 27117833) or renal cell carcinoma (PMID: 25538263) varies between 25% and 50%.

We reworded our sentence: « Approximately 20% of HCCs and BTCs are theoretically good candidates for ICIs based on their high level of expression of programmed death-ligand 1 (PD-L1) and tumor-infiltrating lymphocytes (TILs)  [15,16]. »

ICI monotherapy:

The authors should mention the phase II Checkmate 040 study, based on which nivolumab was approved for second-line treatment in the US.

Thanks, we added the results of this study in the Table 1 and the following text:

« Nivolumab was recently approved for second-line treatment in the United States based on the phase I/II Checkmate 040 study in advanced HCC patients. »

Combination of ICIs:

The authors should mention that the Checkmate 040 was a multi-cohort study, and that the combination of nivolumab plus ipilimumab was tested in one cohort of these cohorts.

Thanks, we added the following text:

« The Checkmate-040 trial was a randomized multi-cohort phase II study assessing nivolumab plus ipilimumab in one of the cohorts at different dosing regimens in 148 pre-treated and naive patients [38]. »

Combinations of angiogenesis inhibitors and ICIs:

45: an update was presented at ESMO 2019 including 67 patients. Please update.

Thanks for this comment. We updated the reference and the Table 1.

The authors should discuss the impact of the positive IMbrave150 study on the future management of HCC.

Thank you for your comment. We added the following text:

«The results of this positive study are a breakthrough, setting the combination of ICI and angiogenesis inhibitors as a future standard of care in the management of HCC, although the high cost which may limit access to these drugs. »

Others:

The authors should include a Table showing ongoing phase I-III combination trials in HCC, categorized according to combination partner (e.g., ICI, anti-angiogenic agent, anti-TGF-b, etc.)

We added a new Table 2: Ongoing combination trials evaluating immune therapies in hepatocellular carcinoma.

Some typos/grammar errors need to be corrected throughout the paper.

We apologize for these errors, the manuscript has been edited and the errors were carefully corrected throughout the text.

The authors should include a paragraph on ‘Future perspectives’

Thanks, we have change the Conclusion paragraph for “Conclusion and future perspectives”

“HCCs and BTCs share a common unique microenvironment immune niche (i.e. the liver).

HCCs are immunogenic but immunosuppressed and highly vascularized tumors. ICI monotherapies were disappointing, but recent results of combinations with antiangiogenics are very encouraging.

BTCs are an anatomically, molecularly and therapeutically heterogeneous group of tumors, with a poor prognosis. Immunotherapies are under clinical development in BTC, and show activity in an ill-defined subgroup of immunogenic tumors, extending beyond MSI phenotype.

Most promising results come from approaches combining targeted therapy on one side and immune therapy to restore or boost the function of immune cells on another side, not only in liver cancers but also in other tumor types. For example, in advanced gastric or colorectal cancers, the combination of regorafenib with nivolumab was tested with interesting activity, not limited to MSI patients [109]. Such combinations would warrant evaluation in HCC and BTC. However, they are associated with increased toxicity; optimization of the handling and schedule of these agents (e.g.dose adjustment, sequential administration) may improve their tolerance.

In both HCCs and BTCs, only a subset of patients benefits from ICIs. Classical biomarkers (e.g. mutation burden, PD-L1 expression, TILs) display variable sensitivity and specificity across tumor types. Response to ICIs may be modulated by the microenvironment immune pattern, which is currently extensively explored as a source of predictive biomarkers. Intratumoral heterogeneity is a challenge for this type of biomarker, particularly in the advanced setting, where limited quantities of tumor material (biopsies) are available. Liquid biopsies (e.g. circulating DNA, RNA, vesicles, tumor cells) may provide useful information but are so far mainly prognostic rather than predictive markers, and not validated yet for routine practice. Overall, the process of companion biomarker identification is a continuous, bench-to-bedside and bedside-to-bench process.

Future progress will come from clinical trials evaluating rationale combinations, and including extensive translational research on blood and tumor to better understand the mechanisms of response or resistance to immune therapy and guide patient selection for these approaches.”

Reviewer 2 Report

The manuscript is a comprehensive review of new immune therapies for liver cancer. It is well written and well understood. There are some issues to be considered:
1) Some genes are not written in italics. Some abbreviations are difficult to follow when reading the article. Those that appear a few times in the text can be deleted.
2) Are there any studies on immune therapy for hepatoblastoma?
3) Vaccines are not treated just in the review, although they are currently relatively few compared to those studying cellular therapies and checkpoint inhibitors. The current status of peptide vaccines, oncolytic viruses, and dendritic cells may be briefly commented on.

Author Response

Reviewer #2:

The manuscript is a comprehensive review of new immune therapies for liver cancer. It is well written and well understood. There are some issues to be considered:

1) Some genes are not written in italics. Some abbreviations are difficult to follow when reading the article. Those that appear a few times in the text can be deleted.

This has been corrected.

2) Are there any studies on immune therapy for hepatoblastoma?

We checked all the 51 clinical trials for hepatoblastoma referenced on clinicaltrials.gov and found two studies evaluating CAR T cells (NCT03618381, NCT04093648). Results are not available yet. We added this precision in the new paragraph on adoptive cell therapies:

« Moreover, clinical trials of CAR-T-cells for solid tumors have been recently launched after proving efficacy in hematologic diseases [43]. Their clinical development in HCC and hepatoblastoma followed promising results from pre-clinical studies and mostly involves AFP, glypican-3, epithelial growth factor receptor and HBV antigens (NCT03618381; NCT04093648) [44].  »

3) Vaccines are not treated just in the review, although they are currently relatively few compared to those studying cellular therapies and checkpoint inhibitors. The current status of peptide vaccines, oncolytic viruses, and dendritic cells may be briefly commented on.

Thank you for this suggestion. We added the following paragraph:

« Furthermore, other modalities of immunotherapy are under evaluation in HCC, such as peptide and dendritic cell-based vaccines, oncolytic viruses and chimeric antigen receptor (CAR) T-cell therapies. Peptide vaccines based on HCC-associated antigens (AFP, glypican-3, telomerase reverse transcriptase) have shown limited efficacy although inducing T cell specific responses [40]. Dendritic cell-based vaccines have been evaluated in many small studies in HCC with encouraging results providing hope for further development [41]. Besides, early-phase trials evaluating adenovirus and vaccinia virus in HCC patients, that selectively target and replicate in cancer cells, reported objective tumor responses [42]. A randomized phase III study is ongoing evaluating the combination of vaccinia virus with sorafenib (NCT02562755). Moreover, clinical trials of CAR-T-cells for solid tumors have been recently launched after proving efficacy in hematologic diseases [43]. Their clinical development in HCC and hepatoblastoma followed promising results from pre-clinical studies and mostly involves AFP, glypican-3, epithelial growth factor receptor and HBV antigens (NCT03618381; NCT04093648) [44]. However, preliminary trials in HCC showed modest anti-tumor responses [44,45]. Strategies such as identifying more immunogenic antigens, improving T cells tumor penetration, and combining CAR-T cells with other antitumor drugs are explored to improve the efficacy of these immune strategies in HCC and reduce the risk of fatal adverse events (e.g. cytokine release syndrome) [44]. Finally, combining ICI with cellular therapies or other immunotherapeutic approaches (i.e. cancer vaccines and oncolytic viruses) might be synergistic and improve OS of HCC patients (NCT03071094) [46].”

Reviewer 3 Report

Dear authors,

This paper shows current immune therapy for HCC and BTC. It doesn’t describe only clinical results of ICI, but also the rationale of this therapy and some problems to be solved at details. Your manuscript is well organized.

I have some questions and recommend some minor changes.

In introduction, you described that HCC is the second leading cause of cancer. I think liver cancer including HCC, BTC and others is the second in 2012. The second leading cause of cancer only for HCC is not so high. You can find current data from some papers. Or revise this sentence for reasonable introduction. In rationale for using ICI in HCC, it is described that patients with high TMB are responsive to ICIs. (Maybe this is for other cancers.) But 1% of patients are TMB high, and suggesting TMB was not a robust biomarker. These sentences should be deleted or more simplified because TMB has no evidence to use ICI. I recommend to add some comments about CART therapy very shortly, because your title is Immune therapy for liver cancer. Some clinical trials using CART therapy for HCC exists. However, you don’t need to describe it in detail. To describe CART therapy in this paper in detail would lead to be complicated. I just recommend to write a few sentence in introduction about existence of CATR therapy.

Author Response

Reviewer #3:

Dear authors,

This paper shows current immune therapy for HCC and BTC. It doesn’t describe only clinical results of ICI, but also the rationale of this therapy and some problems to be solved at details. Your manuscript is well organized.

I have some questions and recommend some minor changes.

In introduction, you described that HCC is the second leading cause of cancer. I think liver cancer including HCC, BTC and others is the second in 2012. The second leading cause of cancer only for HCC is not so high. You can find current data from some papers. Or revise this sentence for reasonable introduction.

Thank you for this remark. HCC is the third leading cause of cancer death worldwide. We updated the reference and the main text.

« Hepatocellular carcinoma (HCC) is the most frequent primary liver cancer (65,000 and 42,000 new cases/year in Europe and the United States, respectively) and the third leading cause of cancer death worldwide [1]. »

In rationale for using ICI in HCC, it is described that patients with high TMB are responsive to ICIs. (Maybe this is for other cancers.) But 1% of patients are TMB high, and suggesting TMB was not a robust biomarker. These sentences should be deleted or more simplified because TMB has no evidence to use ICI.

These sentences have been deleted.

I recommend to add some comments about CART therapy very shortly, because your title is Immune therapy for liver cancer. Some clinical trials using CART therapy for HCC exists. However, you don’t need to describe it in detail. To describe CART therapy in this paper in detail would lead to be complicated. I just recommend to write a few sentence in introduction about existence of CATR therapy.

Thanks for this interesting suggestion. We added the following part in the new paragraph on other immunotherapeutic approaches:

« Moreover, clinical trials of CAR-T-cells for solid tumors have been recently launched after proving efficacy in hematologic diseases [43]. Their clinical development in HCC and hepatoblastoma followed promising results from pre-clinical studies and mostly involves AFP, glypican-3, epithelial growth factor receptor and HBV antigens (NCT03618381; NCT04093648) [44]. However, preliminary trials in HCC showed modest anti-tumor responses [44,45]. Strategies such as identifying more immunogenic antigens, improving T cells tumor penetration, and combining CAR-T cells with other antitumor drugs are explored to improve the efficacy of these immune strategies in HCC and reduce the risk of fatal adverse events (e.g. cytokine release syndrome) [44]. Finally, combining ICI with cellular therapies or other immunotherapeutic approaches (i.e. cancer vaccines and oncolytic viruses) might be synergistic and improve OS of HCC patients (NCT03071094) [46]. »

We hope that these revisions improve the manuscript such that you and the Reviewers now deem it suitable for publication in Cancers.

Once again, we thank you for your interest in our work and we are honoured that you have accepted to consider this review for publication in Cancers.

We remain at your disposal and we look forward to hearing from you at your earliest convenience,

Best regards,

Dr. Cindy Neuzillet

Round 2

Reviewer 1 Report

The authors revised the paper almost satisfactorily. Only table 2 needs some more attention. The list of studies included is incomplete, important studies are missing, i.e. the phase III combination of atezolizumab plus cabozantinib (NCT03755791) or the phase III combination of nivolumab and ipilimumab (NCT04039607). Please check again for ongoing trials on clinical trials.gov and update this table.

Author Response

The authors revised the paper almost satisfactorily. Only table 2 needs some more attention. The list of studies included is incomplete, important studies are missing, i.e. the phase III combination of atezolizumab plus cabozantinib (NCT03755791) or the phase III combination of nivolumab and ipilimumab (NCT04039607). Please check again for ongoing trials on clinical trials.gov and update this table.

Thank you for this comment. ClinicalTrials.gov has been carefully checked and the table has been updated.